# Application of Macrophytes to the Assessment and Classification of Ecological Status above and below the Barrage with Hydroelectric Buildings

**Paweł Tomczyk \*, Mirosław Wiatkowski \* and Łukasz Gruss \***

Institute of Environmental Engineering, Wrocław University of Environmental and Life Sciences, Grunwaldzki Square 24, 50-363 Wrocław, Poland

\* Correspondence: pawel.tomczyk@upwr.edu.pl (P.T.); miroslaw.wiatkowski@upwr.edu.pl (M.W.); lukasz.gruss@upwr.edu.pl (Ł.G.);
Tel.: +48-71-320-5547 (P.T.); +48-71-320-5185 (M.W.); +48-71-320-5547 (Ł.G.)

**Abstract:** The key goal of the Water Framework Directive is to achieve a good ecological status in water bodies. The ecological status is mainly determined by the biological elements, which are a very good indicator of the changes taking place in water environments. Thus, this article focuses on the analysis of different methods of assessment of the ecological status of water bodies based on macrophytes used in selected countries in the European Union (the Macrophyte Index for Rivers (MMOR)—Poland; the Mean Trophic Rank (MTR)—Ireland; the Trophic Index of Macrophytes (TIM)—Bavaria, Germany; the Bulgarian Reference Index of Macrophytes (RI-BG)—Bulgaria). Three research sections have been selected for research on the river Ślęza: The reference section, the section above the barrage and the section below the barrage. The analysis carried out revealed considerable similarity between the results obtained by all these methods—the differences were at most by one class of ecological status (and the analysis of sums of Wilcoxon's ranks revealed that there were no differences between the results obtained using different methods, i.e., p = 0.860). With respect to surface waters, investigation of biological elements is important because it allows one to retrace the past and foresee the future based on the past and present trends in the changes occurring in the species diversity and structure of not only macrophytes, but also other groups of organisms. Further action is required that would determine the scope of influence of barrages with hydroelectric buildings on the environment (in the case of the investigated barrage this influence is negative).

**Keywords:** hydroelectric buildings; macrophytes; ecological status; water quality; rivers

## 1. Introduction

In the EU, the approach to the quality assessment of water resources has been gradually changing over the last dozen or so years. At present, water is considered a heritage that requires protection, it cannot be seen as a commercial product—as can be read in the preamble to the Water Framework Directive [1]. Currently, there is much talk about the so-called sustainable water management, which is echoed in the primary objective of the Directive, i.e., the achievement of good water status (ideally by 2015, otherwise within subsequent 6-year implementation periods, i.e., by 2021, 2027 and so on). Good water status shows by good ecological status, hereafter denoted ES (biological, physicochemical and hydromorphological). Particular focus is on the biological elements; thus, in this case, the key role is played by the preservation of environment, thanks to which valuable habitats and flora and fauna species are protected. This issue is regulated, among others, by the Habitats Directive and the Birds Directive [1–6].

This article is concerned with an assessment of the ecological status (ES) of biological elements (macrophytes), which are the main components accounted for when evaluating ES in surface water bodies. The article assesses the influence of a barrage with hydroelectric buildings located on a lowland river, the Ślęza (a tributary to the Odra), in the southwestern part of Poland, onto the species diversity and structure of occurrence of macrophytes.

In Poland, the Macrophyte Index for Rivers is currently in use. This method evaluates the degree of river habitat degradation (advancement of eutrophication, expressed by trophic indicators) based on the assessment of plant taxa occurring in selected types of macrophyte rivers (selected based on local conditions). The method complies with the provisions of the Water Framework Directive. It should be noted that this group of methods is also used in other EU countries, e.g., Ireland (Mean Trophic Rank = MTR), Germany (Macrophyte Trophic Index = TIM), Great Britain (River LEAFPACS2) or France (Macrophytes Index for Rivers = IBMR) [7–13]. These methods are constantly updated and the current trend is to unify the classification procedures for water bodies. The assessment is carried out within the so-called ecoregions with specific environmental conditions. When interpreting the results, other properties of water bodies are often taken into account: Biological, hydrological, morphological or physicochemical [12].

Another group of methods is based on the investigation of acidity in rivers. This approach has been proposed by French researchers and was first used in Germany. The method consists of dividing the watercourse into homogeneous sections, mapping the vegetation, identifying the species and their distribution and determining the relationship between the water quality parameters and the occurrence of species. Eventually, a floral and ecological map is created and a classification of watercourses is proposed in terms of their acidity. However, in contrast to the methods mentioned above, the methods of this type have not been implemented to the monitoring under the Water Framework Directive [14].

This article interprets the results obtained using the Polish method of macrophyte assessment and relates them to the aforementioned assessments used in the EU. The following countries are selected for comparison: Ireland (the MTR method), Germany (the TIM method) and Bulgaria (the RI-BG method).

## 2. Survey of Literature on the Subject

Macrophytes or aquatic plants are an important indicator of the dynamics of ES of water bodies: They indicate long-term trends in the changes in this status. The influence of river barrages with hydroelectric buildings on the well-being, status and population of macrophytes is discussed in the literature too often. However, publications on the hydropower engineering in Poland and Europe are well known, including those on the operation of such structures [15,16]. Moreover, researchers relate the macrophytes mainly to the research carried out on water reservoirs and their operation as ecosystems, but also in the context of rational water management in such structures [17–19]. Additionally, research on macrophytes is often undertaken when carrying out river inventorying; however, it is done without relating to the existing hydroelectric buildings, only as part of monitoring of water quality in surface water bodies [20,21]. Large scale water management undertakings are an exception. This is because their influence on the environment and the economical development and quality of human life is very high. An example is provided by the Three Gorges Dam in China, which has been and still is comprehensively investigated due to the huge cost of construction, operation and maintenance and the sweeping consequences for both the inhabitants and the environment [22].

The impact on the macrophytes is ambiguous—research exists which suggests that the number of macrophyte taxa below barrages with hydroelectric buildings increases and that their value in terms of quality is also on the rise; yet other research indicates that the macrophyte species become impoverished and that some population structures disappear. Results of research vary depending on the structure under study, i.e., the scope of transformation varies for different damming heights and water level differences resulting from the operation of a hydropower plant. It is assumed that when the difference of water levels is less than 7 m, the influence is imperceptible or even positive, but for higher differences it becomes negative. This influence applies mainly to hydropower plants with

reservoirs, in which the water level changes more than in rivers, where run-of-the-river stations are used. Species impoverishment affects macrophyte species which occurs at small or average depths; the species which prefer deep habitats are not affected. In reservoirs influenced by hydroelectric buildings the number of ruderal species increases and the number of stress-tolerant and competitive species diminishes. This means that in such waters human influence is visible, ruderal plants are flourishing in highly transformed areas, which are often urbanized [23–26].

The prior published research that was carried out reveals both the positive and the negative influence of hydroelectric buildings (hydropower plants) on macrophytes—in this case, the overall plant cover and species diversity and richness within the hydroelectric buildings were investigated. Species equality did not show any correlation; however, slight dominance of *Leptodictyum riparium* and *Veronica anagallis-aquatica* was observed in the watercourses under study. These are rush plants, which grow on watercourse banks. They are resistant to different habitat conditions and cannot be seen as valuable, reference taxa for a given watercourse section because they can occur in various environmental conditions and have no distinct value as pollution indicators (they occur both in highly polluted and clean environments). Consequently, a slightly negative, although not very significant influence of hydroelectric buildings on the living conditions of macrophytes can be seen from this research [27–29].

## 3. Study Area

Our research was carried out in field conditions. Three 100 m-long sections on the river Ślęza were selected—the first, reference section was selected 18.5 km upstream the barrage (near Rzeplin, a small village in the commune of Żórawina), the second one above the hydroelectric buildings (Small Hydropower Plant—SHP) in Wrocław and the third one below the SHP. The research sections are located as follows: The reference section—km 21 + 450–21 + 550 of the river Ślęza, the section upstream the hydropower plant—km 3 + 020–3 + 120, the section downstream the hydropower plant—km 2 + 900–3 + 000.

The location of research sections is shown in Figure 1 below.

**Figure 1.** The location of the research sections in the Macrophyte Index for Rivers MMOR method—above and below the small hydropower plant (SHP) Ślęza (including the reference section).

## 4. Research Methods

### 4.1. The MMOR Method (Poland)

The MMOR method is based on the assessment of macrophytes in 100 m-long watercourse sections. Following the selection of sections, described above, the next step was to walk along the section upstream and to identify the species of water plants (underwater, floating and emergent). Finally, on the way back (downstream), a more general assessment was made in terms of morphology of the channel and the adjacent land as well as the degree of cover with plant species. All the

information was put down on a special form, in which the main criterion is the assessment of trophicity of a given watercourse section. As a final result, the value of the Macrophyte River Index was obtained (after the points assigned to individual indicator species multiplied by their degree of cover were summed up) and allotment to ES class—in Poland it is assessed on a five-level scale, from very good (1) to poor ES (5) for each macrophyte type of watercourse. For highly transformed or artificial areas one can speak of the ecological potential, in line with the provisions of the Water Framework Directive [1,13,30].

The assessment of the ecological element was performed during the growing season, in May 2017, and focused on the river Ślęza in Wrocław, which meets the requirements of the MMOR method (including the presence of aquatic vegetation, shallow and not very wide riverbed and the possibility of exact identification of indicator species). A much wider scope of research was planned, but other rivers did not meet the criteria imposed on them—the river Bystrzyca has no vegetation, despite its natural sections, besides its depth is too high and the Odra in Wrocław is a highly transformed river and its channel has no natural characteristics—banks and bed are made of artificial materials on which vegetation cannot grow [13,30].

As described above and in line with the regulations in force (on rational water management), the most important elements for the ecological assessment are the biological ones. This is because vegetation reacts to changes in the environment, hence the taxonomic composition being assessed in given watercourse sections is a reflection of the aforementioned changes in the environment. Additionally, when recording trends in the changes in vegetation one may draw conclusions on the differences in the condition of environment in the past and at present, which allows one to forecast the possible changes in aquatic environment (for example in terms of reaction or content of a given element in water) [13,30]. The results of field inventorying are given in Tables 2–4.

In all the investigated cases the watercourse under study was classified as abiotic no 19, i.e., sand-and-clay river, which means that the border values of the Macrophyte River Index are identical (class I ≥ 46.8, class II ≥ 36.6, class III ≥ 26.4, class IV ≥ 16.1 and class V < 16.1).

The Macrophyte River Index (MIR), which is the key indicator of the method, is calculated using the following formula [13,30]:

$$\text{MIR} = \frac{\Sigma(P \times L \times W)}{\Sigma(P \times W)} \times 10$$

where:
- -P—taxon cover scale (values from 1–9);
- -L—taxon indicator number (ranging from 1–9);
- -W—taxon weight coefficient (depends on its ecological tolerance—from 1–13).

### 4.2. The MTR Method (Ireland)

The Irish method, the MTR, is based on similar assumptions as the Polish one—each taxon is assigned an appropriate value which determines the macrophyte surface cover (ranging from 1–10; SCV) and the indicator number, which depends on the ecological tolerance of taxa (values from 1–10; STR). The product of these two values yields the overall cover of the area with all the species (CVS). The most important, final result of the procedure is the calculation of the Mean Trophic Rank (MTR), expressed as a value from 10–100. This value is calculated from the following formula (notation as above) [7]:

$$\text{MTR} = \frac{\Sigma CVS}{\Sigma SCV} \times 10$$

In this method research is carried out on 100 m-long sections, by assessing individual taxa. The cover intervals coincide with those in the Polish method, but the indicator numbers have slightly different values for individual taxa. Similar to Poland, in Ireland the macrophyte types of rivers are distinguished, which differ by their characteristics. In line with the classification according to MTR, all the research sections are of type I, i.e., lowland rivers with minimum slope and soft bed, medium prone to eutrophication. The class intervals of ES with respect to macrophytes are the same as those

in sand-and-clay rivers according to the Polish classification (class I ≥ 46.8, class II ≥ 36.6, class III ≥ 26.4, class IV ≥ 16.1 and class V < 16.1) [7].

*4.3. The TIM Method (Germany)*

The TIM method, used in Bavaria (and in this article referred to as the German method for simplicity), is much more complex than the above described MMOR and MTR. The concentration of reactive phosphorus, which is taken up by macrophytes, is assessed (assessment is carried out on 100 m-long watercourse sections). Based on this, the value of the Trophic Index of Macrophytes (TIM) is assessed on the scale from 1.00–4.00, according to the classification by trophic state (from oligotrophic to politrophic; see Table 1 for details). Results are presented on a seven-level scale and have been transposed to the five-level classification as per the Water Framework Directive [9,12,31].

The first step is to map the macrophytes, i.e., determine their distribution in the investigated section. Next, the content of reactive phosphorus is determined, per obtained dry mass of individual taxa (the concentrations are calculated according to literature, accounting for the obtained minimum and maximum concentrations of reactive phosphorus on the investigated section). The fundamental formula that allows one to calculate this value is as follows [31]:

$$PSW = w + x \times s$$

where:

-PSW—the concentration of reactive phosphorus integrated into the plant tissues in the investigated area ($\mu g/dm^3$);

-w—the maximum recorded concentration of reactive phosphorus ($\mu g/dm^3$);

-s—the minimum recorded concentration of reactive phosphorus ($\mu g/dm^3$);

-x—a value based on the relationship between w, s and P (the percentage share of roots in the absorption of phosphorus—with respect to each species), i.e., $x = \frac{P \times w}{(100 - P) \times s}$ .

The final value of TIM is calculated from the following formula [31]:

$$TIM = \frac{\Sigma(IV \times W \times Q)}{\Sigma(W \times Q)}$$

where:

-IV—the value of species as an indicator ($IV = \frac{\Sigma P \times T}{\Sigma P}$; T—the trophic value of species on a scale from 1–4, i.e., from oligotrophy to politrophy, with a step of 0.5 point = 8 trophic states);

-W—weight indicator for each species (depending on the species' tolerance);

-Q—frequency of occurrence of the species in a given watercourse section.

**Table 1.** Classification of ES according to TIM (Germany and Water Framework Directive) [31].

| Germany (TIM) | | | Water Framework Directive (WFD) | | |
|---|---|---|---|---|---|
| Class | Number of Points | Class Description (Trophic State) | Class | Number of Points | Class Description (ES) |
| I | 1.00–1.44 | oligotrophic | I | 1.00–1.59 | very good |
| II | 1.45–1.86 | oligo-mesotrophic | II | 1.60–2.19 | good |
| III | 1.87–2.24 | mesotrophic | III | 2.20–2.79 | moderate |
| IV | 2.25–2.62 | meso-eutrophic | IV | 2.80–3.39 | poor |
| V | 2.63–3.04 | eutrophic | V | 3.40–4.00 | very poor |
| VI | 3.05–3.49 | eu-politrophic | | | |
| VII | 3.50–4.00 | politrophic | | | |

*4.4. The RI-BG Method (Bulgaria)*

The last method in our comparison is the Bulgarian Reference Index of Macrophytes (RI-BG), which is based on the assessment of taxa in three reference groups, depending on the river type. In this case the macrophytes are divided into group A (reference species in a given river type—reference

taxa), group B (the so-called indifferent taxa) and group C (the taxa which cause degradation and do not occur in natural conditions in this type of river—degradation indicators). One can distinguish three main groups of rivers used for assessment in this method (R-G1, R-G2, R-G3). In each of these groups, different macrophyte indicator species are distinguished. The description of types of river groups is given in Table 2. The assessment is carried out on 100 m-long watercourse sections, in which the abundance of macrophyte taxa that occur in it (macrophyte abundance—MA) is assessed based on a 5-class Kohler's scale [32] given in Table 3. Additionally, in this method the quantity (Q) of taxa from a group is determined: This value equals the cubic power of macrophyte volume ($Q = MA^3$). The method also accounts for the structural characteristic of the watercourse channel, flow velocity, bed, shading, as well as the information on the colour and smell of water, if they diverge from standard [10,33,34].

**Table 2.** Division into river types according to the RI-BG method [34].

| Type of River Group | Characteristics of Type of River Group |
|---|---|
| R-G1 | Mountain and semi-mountain river types (R1–R5) |
| R-G2 | Influenced by groundwater river types (R9, R15) |
| R-G3 | Lowland river types (R7, R8, R10–R14, R16) |

**Table 3.** Characteristics of the macrophyte abundance (MA) classes and their conversion to percentages [32].

| Class No | Description of MA class | Conversion to Percentages |
|---|---|---|
| 1 | very rare | $MA \leq 5\%$ |
| 2 | rare | $5 < MA \leq 25\%$ |
| 3 | common | $25 < MA \leq 50\%$ |
| 4 | frequent | $50 < MA \leq 75\%$ |
| 5 | abundant/predominant | $MA > 75\%$ |

The initial reference index (RI) accounts for the ratio of the number of taxa from groups identified in the assessment and calculates it as follows [10,33]:

$$RI = \frac{\sum_{i=1}^{n_A} Q_{Ai} - \sum_{i=1}^{n_C} Q_{Ci}}{\sum_{i=1}^{n_g} Q_{gi}} \times 100$$

where:
- RI—Reference Index;
- $Q_{Ai}$—quantity of the i-th taxon of group A;
- $Q_{Ci}$—quantity of the i-th taxon of group C;
- $Q_{gi}$—quantity of the i-th taxon of all groups;
- $n_A$—total number of taxa in group A;
- $n_C$—total number of taxa in group C;
- $n_g$—total number of taxa in all groups.

When performing calculations using this formula the result ranges from −100 to +100—if only the taxa from group C occur, one obtains the lowest score and if only the taxa from group A occur— the highest. In order to adjust the RI to values from 0 to 1, the Module Macrophyte Assessment (MMP) is calculated, i.e., [10,33]:

$$M_{MP} = \frac{(RI+100) \times 0.5}{100}$$

The assessment is performed in five ES classes (ecological quality ratio—EQR), i.e., on a scale from very good (class I) to bad (class V). The Ślęza belongs to river group R-G3, i.e., lowland rivers, to type R13, i.e., small and medium lowland rivers with sandy and clay bed, with organic sediments, occasionally with gravel bed [10,33]. Classification of ES using the RI-BG method for rivers of type R13 is given in Table 4.

**Table 4.** Classification of ES according to RI-BG for rivers of type R13 [33].

| Class | RI | EQR (R13) | ES | Class Description |
|---|---|---|---|---|
| I | 34 to 100 | 0.67–1.00 | Very good | The taxonomic composition corresponds totally or nearly totally to undisturbed conditions; no detectable changes in the average macrophyte abundance. |
| II | −4 to 34 | 0.48–0.66 | Good | Slight changes in the composition and abundance of macrophyte taxa compared to the type-specific communities. |
| III | −44 to −5 | 0.28–0.47 | Moderate | The composition of macrophyte taxa differs moderately from the type-specific communities, the communities are significantly more distorted than those observed at good quality; moderate changes in the average macrophyte abundance. |
| IV | −100 to −45 | 0.00–0.27 | Poor | Macrophyte communities deviate substantially from those normally associated with the surface water body type under undisturbed conditions. |
| V | No macrophytes | - | Bad | Large portions of the relevant biological communities normally associated with the surface water body under undisturbed conditions are absent. |

## 5. Results—The MMOR Method

From the calculated Macrophyte River Index formula, after a field survey and comparison of individual plant species with the key, the value of MIR was obtained (complementary information on the basic hydromorphological conditions was also obtained, which can be found in Table 5), which is as follows: For the reference section—36.67, for the section above the barrage on the Ślęza—23.33, for the section below the barrage on the Ślęza—32.73 (Tables 6–8, Figure 2). Consequently, ES of these sections can be classified as good, poor and moderate, respectively. At this stage one may conclude that hydroelectric buildings contributed to the improvement of living conditions of macrophytes and to the appearance of species with higher environmental requirements and, consequently, less ecological tolerance. However, it should be added that the ecological condition on the section below the SHP is still worse than at the reference point, which is quasi-natural at this section of the Ślęza. This is most probably caused by the visible influence of the city of Wrocław—i.e., the discharges of waste water, the runoff of fuels from the nearby transportation routes or the waste thrown away to water. Moreover, the number of taxa and the worsening of their quality is also caused by the surfaces being more tight—strengthened near the bridges and hydrotechnical structures or profiled at some sections of the watercourse [35–37]. Details on the identified macrophyte taxa and their occurrence in a given type of habitat (and, consequently, information on the pollution of water environment) are given further on in this chapter.

### 5.1. Reference Section (km 21 + 450–21 + 550 of the River Ślęza)

The highest variety of species has been recorded on the natural section of the river, near the village Rzeplin. This section is not artificially transformed, the bed and banks are not strengthened and no influence of built-up areas is recorded—the vicinity of the river consists of meadows and forests and the only threat is agricultural runoff from fields. As can be seen in Table 2 and in Figure 2, 16 macrophyte species were identified, 12 of these were indicator species, accounted for in the calculation of the Macrophyte River Index.

*Potamogeton gramineus*, which covers from 5–10% of the channel bed (it is an underwater plant), is the most valuable of all the species, having the highest indicator number (7 out of 9). It grows in eutrophic, stagnant or slowly flowing water, up to the depth of 3 m; however, it grows best at the depth from 0.5–1.5 m. Moreover, it is resistant to variations of water table and can be found most often in reservoirs.

The following two species are equally valuable (indicator number 6): *Iris pseudacorus* (standing and flowing water, on the banks, within 30 cm; it also occurs on wetlands, bogs, marshes and wet meadows; most often forms small clusters or grows alone) and *Eleocharis palustris* (characteristics similar to those of *Iris pseudacorus*). Both these species are emergent plants and occupy from 1.0–2.5% and from 0.1–1.0% of the channel, respectively. Both these species have a narrow range of ecological tolerance. All of the aforementioned species indicate a good status of aquatic environment on the investigated section.

Other species have exactly opposite properties: *Cladophora* (grows in highly eutrophic places attaching itself to other plants or to the bed), lesser duckweed (*Lemna minor*) (grows in eutrophic water, usually standing, with neutral or basic reaction and is resistant to pollution), greater duckweed (*Spirodela polyrhiza*) (similar to lesser duckweed, with which it forms clusters) and sweet flag (*Acorus calamus*) (standing, eutrophic, shallow water, sandy and silty bed)—their indicator numbers are: 1 for *Cladophora*, 2 for other species. The first three of these species have a moderate range of ecological tolerance and the last one has a broad tolerance.

The species not accounted for in the calculation of MIR have no value in terms of ES assessment because they are too common and have no valuable properties. These include *Phragmites australis*, which occurs in practically all the aquatic environments, including those which are highly transformed by men (it often forms single species fields on large areas—in the case of this section, the species covers 25–50% of bed, grows on banks up to the depth of 1 m, in eutrophic and standing water with reaction from neutral to basic), gypsywort (*Lycopus europaeus*), purple loosestrife (*Lythrum salicaria*) and three-lobe beggartick (*Bidens tripartita*) (all of which grow on banks and prefer standing water, usually wet meadows or wetlands). These species can spread across large distances and grow both on land and in aquatic environments.

One should also mention the identified taxa of medium ecological value (indicator number 3–4): In this case *Veronica beccabunga* (banks of flowing water, mesotrophic), *Potamogeton crispus* (stagnant water of alkaline reaction, eutrophic and mesotrophic, up to the depth of 3 m), *Rorippa amphibia* (watercourse banks, silty bed, standing water), *Lemna trisulca* (stagnant water, shallow places, preferably eutrophic water, usually silty bed, resistant to organic pollution, grows on other plants and elements in the channel) and *Alisma plantago-aquatica* (standing and eutrophic water, sandy and silty bed). These species are rather valuable, usually display medium ecological tolerance, hence in our country they are moderately aboundant, more or less common [38–40].

In conclusion, in the course of research on this watercourse section it was found that its flow is calm or there is hardly any flow at all, the water is usually eutrophic, rarely mesotrophic and the reaction varies from neutral to basic. The depth almost never exceeds 1 m. There are wet meadows on the banks and occasionally wetlands resembling lowland bogs. Because of the macrophyte taxa that occur here, the bed material is predominantly silt or sand, which is also mentioned further on in this chapter, when extra information complementary to the analysis is provided.

## 5.2. Section above the Barrage on the River Ślęza (km 3 + 020–3 + 120)

The least species diversity and quality of species composition and, consequently, the worst ES were recorded on the research section above the barrage on the river Ślęza—as mentioned above, this situation is influenced by the increased share of sealed bed surface area and banks of the channel. Consequently, aquatic plants cannot grow. Another factor is the inflow of pollution of anthropogenic origin—fuels from transportation routes, waste water from households or industry and waste thrown away to water.

In this case only four macrophyte species were recorded, out of which two are not indicator species. The other two are typical of habitats rich in biogens, with strong anthropopression.

The biological diversity is visibly smaller, not only in terms of the species structure, but also in terms of the degree of cover by water plants—only *Phragmites australis*, which grows in the channel, was estimated to cover from 1.0–2.5% of the surface, the second species in terms of the degree of cover is *Lythrum salicaria*—from 0.1–1.0%. It should be noted that neither of these two species is accounted for when calculating MIR. These are common species with broad range of ecological tolerance; however, they grow mainly on watercourse banks. The preferred reaction ranges from neutral to basic. Standing or slowly flowing water is also preferred. Moreover, purple loosestrife (*Lythrum salicaria*) grows mainly on wet meadows and wetlands. It is also worth mentioning that on both banks of the Ślęza above the barrage a tendency for forming single species clusters of *Phragmites australis* was visible—in the flooded areas the cover scale for this monocot was close to 100%.

In terms of the indicator number, the other species have average (*Rorippa amphibia*) or low (*Lemna minor*) importance. The highest numbers of these species can be found in standing or slowly flowing water rich in biogens, often polluted and sometimes silty. Their cover was scarce, one station for each species was recorded—great yellowcress was growing on the right bank at the beginning of the research section, duckweed was found in a similar place, but had a tendency to attach itself to objects existing in the channel—some of which were waste. In the middle of channel no macrophyte taxa were found. Conditions unfavourable for vegetation were the main reason for this—depths exceeding 2 m together with high turbidity prevented the light from penetrating to the deeper layers of water, which in consequence prevented plants from growing there. During hydromorphological research, in some places in the higher parts of the channel river bars were visible, in which scarce macrophytes developed.

Based on the above information one might conclude that the habitat conditions on this section were potentially similar to those recorded at the reference point (standing or slowly flowing water, basic or neutral reaction, water rich in biogens). The only difference was that the visible turbidity and depth were definitely higher. Moreover, there were significant anthropopression factors acting on individual elements of the watercourse.

### 5.3. Section below the Barrage on the River Ślęza (km 2 + 900–3 + 000)

Compared to other research sections, ES of the section below the barrage on the river Ślęza was average—moderate ecological status. In this case we may say that the influence of the barrage on the structure and composition of water plant taxa was positive due to the location of sections directly before and after the barrage, with no other influences.

Although only seven macrophyte species were recorded, the structure of indicator species was so favourable (valuable species with high indicator number) that an almost good status was achieved, with four indicator species. Compared to the section above the barrage, the cover of *Phragmites australis* in this section is higher—from 5.0–10.0%—but the species does not form such vast fields as above the barrage and more plants occur in the channel. There is less *Lythrum salicaria*—below 0.1%, more *Lemna minor*—from 1–2.5% and more *Rorippa amphibia*—from 0.1–1%. Comparing the species which can be found both above and below the barrage, one can conclude that the depths are smaller and there are more dry areas, which are not classified as wetlands. Currently, there is also more wood load and there are more fallen trees below the barrage, compared to the section above it. Consequently, zones of quiet flow are formed, which allows vegetation such as *Lemna minor* to grow, for example, on dead trunks.

The most visible change is in those species, which cannot be found above the barrage—if the appearance of *Lycopus europaeus* which is common in all the lowlands and grows on the banks of quiet water bodies is not surprising (it is not an indicator species and has a broad ecological tolerance), the appearance of *Potamogeton gramineus* in an urbanized, transformed area is quite sensational. The plant has a rather narrow range of ecological tolerance and prefers shallow stations rich in biogens. The requirement of achieving the depth in the range of 0.5–1.0 m above the barrage was impossible to meet; however, below the barrage, the depth of water exceeds 1.0 m on a section of at least 50 m, which allows vegetation to grow thanks to access to light. *Potamogeton gramineus* covers from 25–50%

of the channel, hence, when calculating the Macrophyte River Index, it makes the result significantly higher.

Surprisingly, there is a species which has not been found even in the reference section—namely *Rumex hydrolapathum*, whose indicator number is 4. However, its cover is less than 0.1% (only one station was recorded). This species can be found in shallow mesotrophic water with calm flow or even in stagnant water. Its presence indicates that the concentration of biogens is lower, compared to the section above the barrage.

With reference to this section, it is also worth mentioning that the water is mesotrophic and eutrophic, with lower concentration of biogens than above the barrage and smaller depth, from 0.5–1.0 m. There is a change in the species structure due to the changes in habitat conditions—more light is accessible for aquatic plants. Zones of quiet water are formed thanks to the greater amount of wood load and more fallen trees. The surface area of wetlands is smaller. The taxonomic diversity is higher than above the barrage, but lower than at the reference point. Barrages improve the composition of species structure of vegetation, the increase of oxygen concentration below the barrage stimulates the development of flora.

Complementary information on the basic hydromorphological conditions of the river Ślęza channel on the research sections and on the conditions in which the research was performed can be found in Table 5. Moreover, in Figure 2 the determined number of taxa and the calculated MIR for each section of the river Ślęza are shown.

**Table 5.** Complementary information about the sections selected for the MMOR analysis.

| Bed material | Secondarily covered with silt (majority), sand—the reference section, Rzeplin; sand (predominant), silt—sections in Wrocław |
|---|---|
| Channel width | 1–5 m: Rzeplin—1.5 m (on the average), Wrocław—3.0 m (on the average) |
| Depth | 0.5–1.0 m and > 1.0 m (predominantly 2.0 m) |
| Shading | Rzeplin—partial, Wrocław—no shadow or partial shadow |
| Visibility | Rzeplin—good (water slightly turbid or visible bottom), Wrocław above and below the barrage—poor (high turbidity, which hinders identification of macrophytes) |

**Figure 2.** Number of taxa and MIR in individual research sections, the Ślęza, May 2017.

**Table 6.** Summary of MMOR results, May 2017, reference section on the Ślęza (P—taxon cover scale, P%—percentage taxon cover scale, L—taxon indicator number, W—taxon weight coefficient).

| No | Taxon | P | P% | L | W | P×L×W | P×W | | | | |
|---|---|---|---|---|---|---|---|---|---|---|---|
| 1. | *Cladophora* | 2 | 0.1–1.0 | 1 | 2 | 4 | 4 | | | | |
| 2. | *Lycopus europaeus* | 1 | <0.1 | - | - | | | | | | |
| 3. | *Iris pseudacorus* | 3 | 1.0–2.5 | 6 | 2 | 36 | 6 | | | | |
| 4. | *Lythrum salicaria* | 2 | 0.1–1.0 | - | - | | | | | | |
| 5. | *Eleocharis palustris* | 2 | 0.1–1.0 | 6 | 2 | 24 | 4 | | | | |
| 6. | *Veronica beccabunga* | 2 | 0.1–1.0 | 4 | 1 | 8 | 2 | | | | |
| 7. | *Potamogeton gramineus* | 5 | 5.0–10.0 | 7 | 1 | 35 | 5 | | | | |
| 8. | *Potamogeton crispus* | 7 | 25–50 | 4 | 2 | 56 | 14 | | **MIR** | | **ES** |
| 9. | *Rorippa amphibia* | 1 | <0.1 | 3 | 1 | 3 | 1 | | | | |
| 10. | *Lemna minor* | 4 | 2.5–5.0 | 2 | 2 | 16 | 8 | | | | |
| 11. | *Lemna trisulca* | 3 | 1.0–2.5 | 4 | 2 | 24 | 6 | | | | |
| 12. | *Spirodela polyrhiza* | 4 | 2.5–5.0 | 2 | 2 | 16 | 8 | | | | |
| 13. | *Acorus calamus* | 2 | 0.1–1.0 | 2 | 3 | 12 | 6 | | | | |
| 14. | *Phragmites australis* | 7 | 25.0–50.0 | - | - | | | | | | |
| 15. | *Bidens tripartita* | 2 | 0.1–1.0 | - | - | | | | | | |
| 16. | *Alisma plantago-aquatica* | 1 | <0.1 | 4 | 2 | 8 | 2 | | | | |
| | **SUM** | | | | | **242** | **66** | **36.67** | **II—good** | | |

**Table 7.** Summary of MMOR results, May 2017, section above the barrage on the Ślęza.

| No | Taxon | P | P% | L | W | P×L×W | P×W | | | | |
|---|---|---|---|---|---|---|---|---|---|---|---|
| 1. | *Lythrum salicaria* | 2 | 0.1–1.0 | - | - | | | | | | |
| 2. | *Rorippa amphibia* | 1 | <0.1 | 3 | 1 | 3 | 1 | | **MIR** | | **ES** |
| 3. | *Lemna minor* | 1 | <0.1 | 2 | 2 | 4 | 2 | | | | |
| 4. | *Phragmites australis* | 3 | 1.0–2.5 | - | - | | | | | | |
| | **SUM** | | | | | **7** | **3** | **23.33** | **IV—poor** | | |

**Table 8.** Summary of MMOR results, May 2017, section below the barrage on the Ślęza.

| No | Taxon | P | P% | L | W | P×L×W | P×W | | | | |
|---|---|---|---|---|---|---|---|---|---|---|---|
| 1. | *Lycopus europaeus* | 2 | 0.1 –1.0 | - | - | | | | | | |
| 2. | *Lythrum salicaria* | 1 | <0.1 | - | - | | | | | | |
| 3. | *Potamogeton gramineus* | 2 | 0.1–1.0 | 7 | 1 | 14 | 2 | | **MIR** | | **ES** |
| 4. | *Rorippa amphibia* | 2 | 0.1–1.0 | 3 | 1 | 6 | 2 | | | | |
| 5. | *Lemna minor* | 3 | 1.0–2.5 | 2 | 2 | 12 | 6 | | | | |
| 6. | *Rumex hydrolapathum* | 1 | <0.1 | 4 | 1 | 4 | 1 | | | | |
| 7. | *Phragmites australis* | 5 | 5.0–10.0 | - | - | | | | | | |
| | **SUM** | | | | | **36** | **11** | **32.73** | **III—moderate** | | |

## 6. Comparison of Results with Selected (Macrophyte-Based) Methods of ES Assessment Used in the EU

### 6.1. The MTR Method (Ireland)

Although the results obtained from the MTR method are similar to those obtained in MMOR, the indicators are different. A good status was obtained on the reference section (MTR = 37.56), a poor status above the SHP (26) and a moderate status below the SHP (32.73). The assessment accounted for the following factors (respectively): 12, 3 and 5 taxa, which is different from MMOR (and follows from the fact that in MTR the taxa, for which the indicator number is not determined are not taken into account). These relationships are shown in Figure 3. Detailed results are given in Tables 9–11.

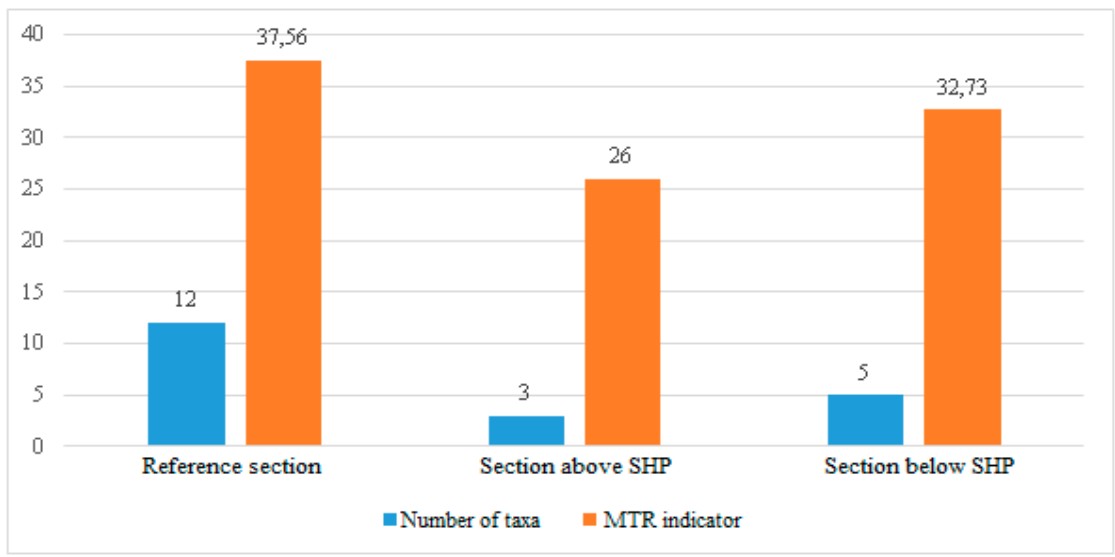

**Figure 3.** Number of taxa and MTR in each research section, the Ślęza, May 2017.

**Table 9.** Summary of MTR results, May 2017, reference section on the Ślęza.

| No | Taxon | SCV% | SCV | STR | CVS | | |
|----|-------|------|-----|-----|-----|----|----|
| 1. | *Cladophora* | 0.1–1.0 | 2 | 1 | 2 | | |
| 2. | *Lycopus europaeus* | <0.1 | 1 | - | - | | |
| 3. | *Iris pseudacorus* | 1.0–2.5 | 3 | 5 | 15 | | |
| 4. | *Lythrum salicaria* | 0.1–1.0 | 2 | - | - | | |
| 5. | *Eleocharis palustris* | 0.1–1.0 | 2 | 7 | 14 | | |
| 6. | *Veronica beccabunga* | 0.1–1.0 | 2 | - | - | | |
| 7. | *Potamogeton gramineus* | 5.0–10.0 | 5 | 7 | 35 | **MTR** | **ES** |
| 8. | *Potamogeton crispus* | 25–50 | 7 | 4 | 28 | | |
| 9. | *Rorippa amphibia* | <0.1 | 1 | 3 | 3 | | |
| 10. | *Lemna minor* | 2.5–5.0 | 4 | 2 | 8 | | |
| 11. | *Lemna trisulca* | 1.0–2.5 | 3 | 4 | 12 | | |
| 12. | *Spirodela polyrhiza* | 2.5–5.0 | 4 | 3 | 12 | | |
| 13. | *Acorus calamus* | 0.1–1.0 | 2 | 4 | 8 | | |
| 14. | *Phragmites australis* | 25.0–50.0 | 7 | 2 | 14 | | |
| 15. | *Bidens tripartita* | 0.1–1.0 | 2 | - | - | | |
| 16. | *Alisma plantago-aquatica* | <0.1 | 1 | 3 | 3 | | |
| | **SUM** | | 41 | - | 154 | **37.56** | **II—good** |

**Table 10.** Summary of MTR results, May 2017, section above the barrage on the Ślęza.

| No | Taxon | SCV% | SCV | STR | CVS | | |
|----|-------|------|-----|-----|-----|----|----|
| 1. | *Lythrum salicaria* | 0.1–1.0 | 2 | - | - | | |
| 2. | *Rorippa amphibia* | <0.1 | 1 | 2 | 2 | **MTR** | **ES** |
| 3. | *Lemna minor* | <0.1 | 1 | 2 | 2 | | |
| 4. | *Phragmites australis* | 1.0–2.5 | 3 | 3 | 9 | | |
| | **SUM** | | 5 | | 13 | **26** | **IV—poor** |

**Table 11.** Summary of MTR results, May 2017, section below the barrage on the Ślęza.

| No | Taxon | SCV% | SCV | STR | CVS | | |
|---|---|---|---|---|---|---|---|
| 1. | *Lycopus europaeus* | 0.1–1.0 | 2 | - | - | | |
| 2. | *Lythrum salicaria* | <0.1 | 1 | - | - | | |
| 3. | *Potamogeton gramineus* | 0.1–1.0 | 2 | 7 | 14 | **MTR** | **ES** |
| 4. | *Rorippa amphibia* | 0.1–1.0 | 2 | 2 | 4 | | |
| 5. | *Lemna minor* | 1.0–2.5 | 3 | 2 | 6 | | |
| 6. | *Rumex hydrolapathum* | <0.1 | 1 | 2 | 2 | | |
| 7. | *Phragmites australis* | 5.0–10.0 | 5 | 2 | 10 | | |
| | **SUM** | | 11 | | 36 | **32.73** | **III—moderate** |

*6.2. The TIM Method (Germany)*

The results obtained (Tables 12–15) indicate that common reed (*Phragmites australis*) has the strongest influence on worsening of all of the results, as it absorbs the highest amount of reactive phosphorus. Despite this, the results are identical to those in the previous methods, i.e., good status on the reference section (mesotrophic), poor status above the barrage (eutrophic) and moderate below the barrage (meso-eutrophic). This means that the more species occur in a watercourse, the better its trophic state is (in this case it is understood to be equivalent to ES).

**Table 12.** Summary of the Trophic Index of Macrophytes (TIM) results, May 2017, reference section on the Ślęza.

| Taxon | s (µg/dm³) | w (µg/dm³) | P (µg/dm³) | x (-) | PSW (µg/dm³) | Trophic classification |
|---|---|---|---|---|---|---|
| *Phragmites australis* | 0.03203 | 0.41633 | | | 432.15 | eutrophic (eu) |
| *Potamogeton crispus* | 0.01441 | 0.18735 | | | 194.47 | meso-eutrophic (m-eu) |
| *Potamogeton gramineus* | 0.00384 | 0.04996 | | | 51.86 | mesotrophic (m) |
| *Spirodela polyrhiza* | 0.00038 | 0.00500 | | | 5.19 | oligotrophic (o) |
| *Lemna minor* | 0.00032 | 0.00416 | | | 4.32 | oligotrophic (o) |
| *Iris pseudacorus* | 0.00102 | 0.01332 | | | 13.83 | oligotrophic (o) |
| *Lemna trisulca* | 0.00013 | 0.00167 | | | 1.73 | oligotrophic (o) |
| *Cladophora* | 0.00001 | 0.00017 | 4.27242 | 0.49427 | 0.17 | oligotrophic (o) |
| *Lythrum salicaria* | 0.00013 | 0.00167 | | | 1.73 | oligotrophic (o) |
| *Eleocharis palustris* | 0.00008 | 0.00100 | | | 1.04 | oligotrophic (o) |
| *Veronica beccabunga* | 0.00006 | 0.00083 | | | 0.86 | oligotrophic (o) |
| *Acorus calamus* | 0.00128 | 0.01665 | | | 17.29 | oligo-mesotrophic (o-m) |
| *Bidens tripartita* | 0.00032 | 0.00416 | | | 4.32 | mesotrophic (m) |
| *Lycopus europaeus* | 0.00001 | 0.00017 | | | 0.17 | oligotrophic (o) |
| *Rorippa amphibia* | 0.00006 | 0.00083 | | | 0.86 | oligotrophic (o) |
| *Alisma plantago-aquatica* | 0.00012 | 0.00150 | | | 1.56 | oligotrophic (o) |

**Table 13.** Summary of the TIM results, May 2017, section above the small hydropower plant (SHP) on the Ślęza.

| Taxon | s (µg/dm³) | w (µg/dm³) | P (µg/dm³) | x (-) | PSW (µg/dm³) | Trophic classification |
|---|---|---|---|---|---|---|
| *Phragmites australis* | 0.02250 | 0.28125 | | | 348.99 | eutrophic (eu) |
| *Lythrum salicaria* | 0.00769 | 0.08345 | 4.92737 | 0.49427 | 87.25 | mesotrophic (m) |
| *Rorippa amphibia* | 0.00020 | 0.00223 | | | 2.33 | oligotrophic (o) |
| *Lemna minor* | 0.00013 | 0.01391 | | | 13.97 | oligotrophic (o) |

**Table 14.** Summary of the TIM results, May 2017, section below the SHP on the Ślęza.

| Taxon | s (μg/dm³) | w (μg/dm³) | P (μg/dm³) | x (-) | PSW (μg/dm³) | Trophic classification |
|---|---|---|---|---|---|---|
| *Phragmites australis* | 0.02250 | 0.28125 | | | 292.37 | eutrophic (eu) |
| *Lythrum salicaria* | 0.00110 | 0.01391 | | | 14.45 | oligotrophic (o) |
| *Rorippa amphibia* | 0.00176 | 0.02225 | | | 23.12 | oligo-mesotrophic (o-m) |
| *Lemna minor* | 0.00216 | 0.02736 | 4.40755 | 0.49427 | 28.43 | oligo-mesotrophic (o-m) |
| *Lycopus europaeus* | 0.00329 | 0.04172 | | | 43.35 | oligo-mesotrophic (o-m) |
| *Potamogeton gramineus* | 0.00527 | 0.06676 | | | 69.36 | mesotrophic (m) |
| *Rumex hydrolapathum* | 0.00165 | 0.00834 | | | 9.16 | oligotrophic (o) |

**Table 15.** ES (macrophytes) as identified by TIM.

| Research Section | TIM | Trophic Classification | Class (TIM) | Class (WFD) | ES (WFD) |
|---|---|---|---|---|---|
| Reference section (Rzeplin) | 2.18 | mesotrophic | III | II | good |
| Section above the SHP (Wrocław) | 2.87 | eutrophic | V | IV | poor |
| Section below the SHP (Wrocław) | 2.46 | meso-eutrophic | IV | III | moderate |

### 6.3. The RI-BG Method (Bulgaria)

Based on the methodological assumptions of the RI-BG method, 7 taxa were selected in the reference section belonging to one of the three groups in the method. For the section above the SHP, 2 taxa were selected and for the section below it—3 taxa. The best results were obtained in the reference section, where ES class I was recorded for the biological elements (EQR = 0.71) and the worst were those for the section above the SHP (IV, EQR = 0.25). The results for the section below the SHP were intermediate (II, EQR = 0.50). This means that the reference section was abundant in reference taxa for this river type and in neutral species, whereas in the section above the SHP species inappropriate for this river type or neutral were predominant. Below the hydrotechnical building an intermediate state was observed. The results of research carried out using the RI-BG method are shown in Tables 16–18.

**Table 16.** Summary of RI-BG results, May 2017, reference section on the Ślęza.

| No. | Taxon | Group | MA | Q | | | |
|---|---|---|---|---|---|---|---|
| 1. | *Cladophora* | - | 1 | 1 | | | |
| 2. | *Lycopus europaeus* | - | 1 | 1 | | | |
| 3. | *Iris pseudacorus* | - | 1 | 1 | | | |
| 4. | *Lythrum salicaria* | B | 1 | 1 | | | |
| 5. | *Eleocharis palustris* | - | 1 | 1 | | | |
| 6. | *Veronica beccabunga* | B | 1 | 1 | | | |
| 7. | *Potamogeton gramineus* | A | 3 | 27 | | | |
| 8. | *Potamogeton crispus* | C | 2 | 8 | **RI** | **EQR** | **ES** |
| 9. | *Rorippa amphibia* | - | 1 | 1 | | | |
| 10. | *Lemna minor* | C | 1 | 1 | | | |
| 11. | *Lemna trisulca* | B | 1 | 1 | | | |
| 12. | *Spirodela polyrhiza* | C | 1 | 1 | | | |
| 13. | *Acorus calamus* | - | 1 | 1 | | | |
| 14. | *Phragmites australis* | - | 3 | 27 | | | |
| 15. | *Bidens tripartita* | - | 1 | 1 | | | |
| 16. | *Alisma plantago-aquatica* | - | 1 | 1 | | | |
| | Overall: | | | | 42.5 | 0.71 | I—very good |

**Table 17.** Summary of RI-BG results, May 2017, section above the SHP on the Ślęza.

| No. | Taxon | Group | MA | Q | | | |
|---|---|---|---|---|---|---|---|
| 1. | *Lythrum salicaria* | B | 1 | 1 | | | |
| 2. | *Rorippa amphibia* | - | 1 | 1 | **RI** | **EQR** | **ES** |
| 3. | *Lemna minor* | C | 1 | 1 | | | |
| 4. | *Phragmites australis* | - | 1 | 1 | | | |
| | Overall: | | | | −50 | 0.25 | IV—poor |

**Table 18.** Summary of RI-BG results, May 2017, section below the SHP on the Ślęza.

| No. | Taxon | Group | MA | Q | | | |
|---|---|---|---|---|---|---|---|
| 1. | *Lycopus europaeus* | - | 1 | 1 | | | |
| 2. | *Lythrum salicaria* | B | 1 | 1 | | | |
| 3. | *Potamogeton gramineus* | A | 1 | 1 | | | |
| 4. | *Rorippa amphibia* | - | 1 | 1 | **RI** | **EQR** | **ES** |
| 5. | *Lemna minor* | C | 1 | 1 | | | |
| 6. | *Rumex hydrolapathum* | - | 1 | 1 | | | |
| 7. | *Phragmites australis* | - | 2 | 8 | | | |
| | Overall: | | | | 0 | 0.50 | II—good |

*6.4. Comparison of Results—The MMOR, MTR, TIM and RI-BG Methods*

As shown in Figure 4, ES assessed by identifying macrophytes using MMOR, MTR, TIM and RI-BG, reveals a 100% agreement in terms of ES classification on the section above the barrage, as per the Water Framework Directive. The differences were observed only on the reference section and the section below the barrage—the RI-BG method indicated ES class I and II (very good and good status), whereas other methods suggested class II and III (good and moderate status). On the section above the barrage, ES class IV, i.e., poor status, was recorded. Despite this, the results are very similar, which may be the consequence of the fact that the methods focus on similar elements, despite the individual approach of each method, which differs, either a bit more, or less, from others. In the EU methods based on the assessment of the watercourse trophic state are currently in use, i.e., enrichment with biogens is analyzed which, however, should change. Currently, the trend is to make the analyses as broad as possible, so as to have the widest possible research context, accounting for as many factors as possible, thus leading to more reliable results than those obtained by methods focusing on just a few elements. Such evaluations are more broadly used in the assessment of other biological elements, including e.g., zoobenthos, phytoplankton or ichthyofauna. Macrophyte-based research is not as much implemented as the above mentioned, which also should change. The above results show that the operation of a barrage with hydroelectric buildings on the Ślęza improved the growth conditions for macrophytes, compared to the section above the SHP. Despite that, the results are not as favourable as those on the section that most resemble natural conditions of the investigated watercourse. It should be mentioned that the sections above and below the SHP are located in close vicinity of urban sprawl, which also influences the final result. For comparison, the reference section is mostly surrounded by forests and meadows and the nearest settlement, Rzeplin, is a small village with no significant impact on the watercourse. This conclusion is particularly visible when the RI-BG method is considered, as the method indicates a strong predominance of taxa characteristic of small and medium lowland rivers with mainly sandy bottom. In these habitats, alien species constitute an insignificant fraction—in contrast to the section above the SHP on the Ślęza, where the taxa inappropriate for this type of river are much more dominant.

Our analysis reveals that the barrage with hydroelectric buildings has influenced the ES of the lowland river, determined based on the Macrophyte River Assessment Method. Considering the number of identified taxa in the investigated sections, the influence on the biological elements of ES, i.e., the structure and occurrence of individual macrophyte taxa, was, in the case of the barrage on the Ślęza, positive. The poorest result was recorded on the section above the SHP (poor status),

medium result—below the building (moderate status), and the best—at the reference point (good status). As can be seen, the biological conditions have worsened considerably on the section from the reference section to the section above the SHP, which is related to anthropopression in the area where the barrage is located. The taxa occurring in this section indicate a more acid reaction of water and its higher content of biogens. The species which can be found here have a broader ecological tolerance, so are less valuable. Hydroelectric buildings have contributed to the improvement of ES—in this case the status changed from poor to moderate. This change was due to the fact that below the barrage taxa with a higher indicator number were found, more valuable and having a narrower ecological tolerance. Above the barrage many non-indicator species occurred, which suggests that the habitat conditions were not very different from other places. These species are common all over the country and have a high tolerance to pollution, particularly that of organic origin. Despite the improvement of habitat quality below the barrage, the status of this section is still worse than that of the quasi-natural reference section, where almost twice as many taxa were recorded, including some of high ecological value.

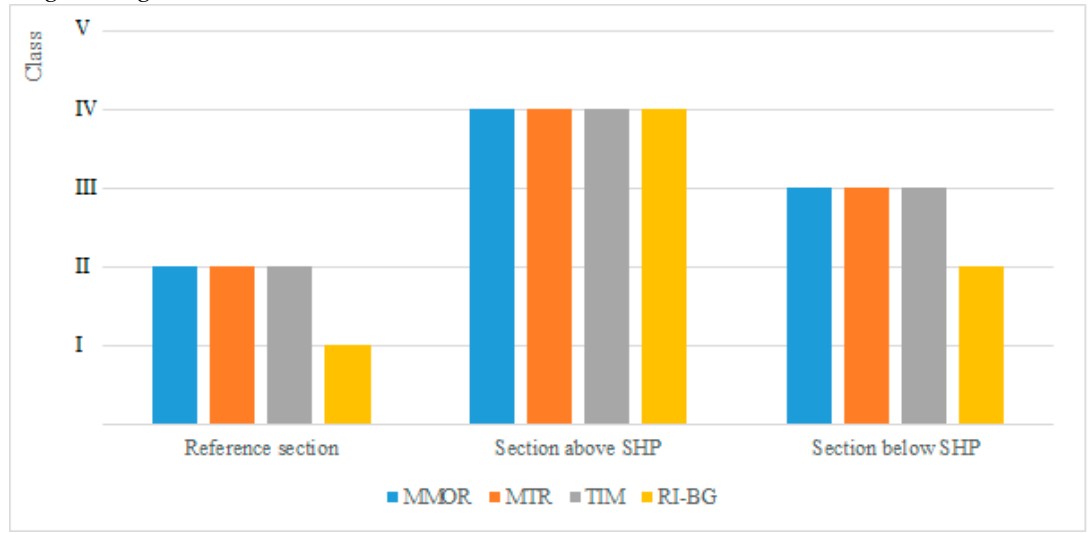

**Figure 4.** Comparison of ES results obtained using MMOR, MTR, TIM and RI-BG.

In order to statistically compare the results obtained by using different methods, the Wilcoxon Rank Sum Test was performed using the SAS University Edition software (Figure 5). So far, this test has been successfully used for statistical assessment of both river and sea water quality monitoring, for the study of physicochemical, hydromorphological and biological elements [41–43].

The null hypothesis assumed that there were no differences between the results obtained from different methods. The hypotheses for the Wilcoxon's test were formulated as follows:

- $H_0 : F_1 = F_2$ (no significant difference in the distributions of variables)
- $H_1 : F_1 \neq F_2$ (the distributions of variables differ significantly)

An analysis of the Wilcoxon's Rank Sum Test (Table 19) revealed that there were no differences between the results obtained using different methods (p = 0.860), which indicates that the null hypothesis should be accepted.

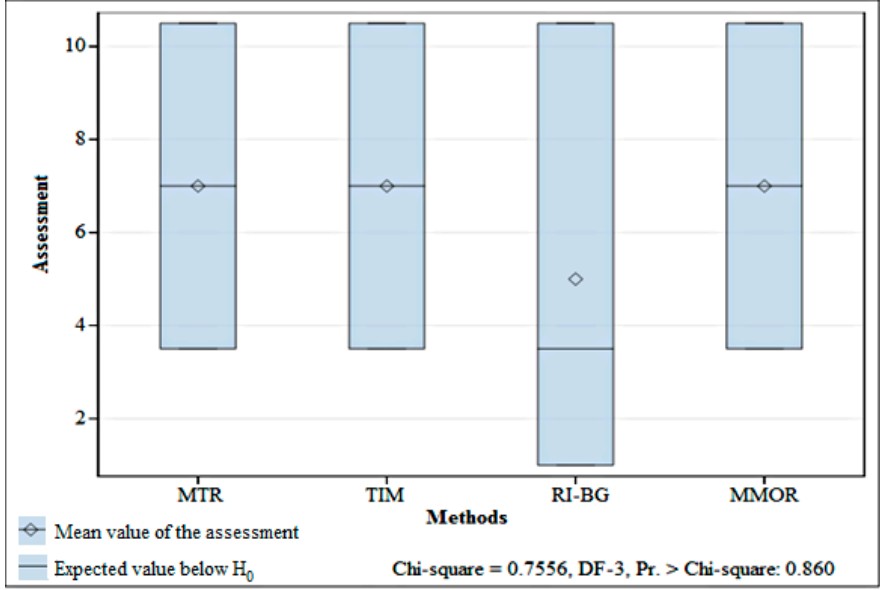

**Figure 5.** Distribution of Wilcoxon's assessment for results obtained using the four methods (Kruskall–Wallis test).

**Table 19.** Wilcoxon's assessment (sums of ranks) for results obtained using the four methods.

| Method | N | Sum of Outcomes | Expected below $H_0$ | Std. dev. below $H_0$ | Average Outcome |
|--------|---|-----------------|----------------------|-----------------------|-----------------|
| MTR    | 3 | 21.0            | 19.50                | 5.176433              | 7.0             |
| TIM    | 3 | 21.0            | 19.50                | 5.176433              | 7.0             |
| RI-BG  | 3 | 15.0            | 19.50                | 5.176433              | 5.0             |
| MMOR   | 3 | 21.0            | 19.50                | 5.176433              | 7.0             |

## 7. Conclusions

Our study marked the positive effects of hydroelectric buildings. All of the comparable macrophyte methodologies proved that the section above the reference location had become a poor water quality over the hydropower station, and the section below the hydropower station had not reached the reference state (but this state was better than in the section above SHP). Therefore, our analysis reveals that the barrage with hydroelectric buildings has influenced the ES of the lowland river, based on the Macrophyte River Assessment Method. Considering the number of identified taxa in the investigation sections, we conclude that in the case of the barrage on the Ślęza, the influence on the biological elements of ES (i.e., the structure and occurrence of individual macrophyte taxa) was positive. The poorest result was recorded on the section above the SHP (poor status), medium result—below the building (moderate status), and the best—at the reference point (good status). As can be seen, the biological conditions have worsened considerably on the section from the reference section to the section above the SHP, which is related to anthropopression in the area where the barrage is located. The taxa occurring in this section indicate a more acid reaction of water and a higher content of biogens. The species found here have a broader ecological tolerance, so are less valuable. Hydroelectric buildings have contributed to the improvement of ES—in this case the status changed from poor to moderate. This change was due to the fact that below the barrage taxa with a higher indicator number, they were found to be more valuable and having a narrower ecological tolerance. Above the barrage many non-indicator species were present, which suggests that the habitat conditions were not very different from those in other places. These species are common all over the country and have a high tolerance to pollution, particularly that of organic origin. Despite the improvement of habitat quality below the barrage, the status of this section is still worse than that

of the quasi-natural reference section, where almost twice as many taxa were recorded, including some of high ecological value.

The discussion of results indicates that in the first case, the MMOR, MTR, TIM and RI-BG methods all lead to similar results. The first three methods produce exactly the same result. The results obtained from the last method are one ES class lower than the others or the same (an analysis of the sums of Wilcoxon's ranks has proved that there is no difference between the results obtained by using different methods (p = 0.860), which indicates that the null hypothesis should be accepted). However, one should note that if more observations were taken, in a larger time horizon, on more research sections and on rivers of various types, the results might be considerably different. All the macrophyte-based assessment methods in the EU attempt to approach the problem comprehensively, but focus on the evaluation of the trophic state, which is the easiest to identify. Using other groups of methods is certainly worth trying, e.g., those based on the evaluation of acidity or alkalinization of environment, which are used when other elements of ES assessment are considered, such as zoobenthos, phytoplankton or ichthyofauna. Thus, one could select a method that gives the most reliable results, which are closest to reality and valid for a long time horizon. Figure 6 shows the step by step methodology for each of the methods discussed in this article.

Investigation of biological elements of water bodies has a future, as it allows one to retrace the past and foresee the future based on past and present trends in the changes to the structure and species diversity of not only macrophytes, but also other groups of organisms [44,45]. Further research is worth pursuing to determine the real scope of influence of barrages with hydroelectric buildings on the environment and determine if it is positive, negative or intermediate.

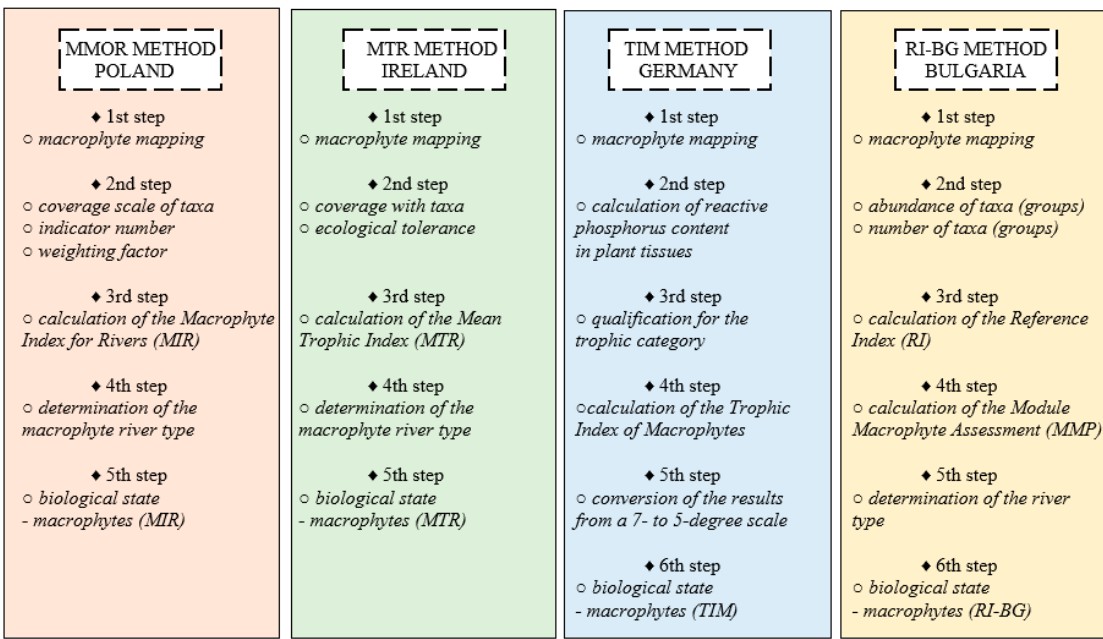

**Figure 6.** Diagram of the (macrophyte-based) ES analysis process for the described methods (own work).

**Author Contributions:** P.T. conceived and designed the experiments; P.T. and Ł.G. analyzed the data; P.T. contributed analysis tools; P.T. carried out field research; P.T. and M.W. wrote the paper, M.W. supervised the work on article.

**Funding:** This research received no external funding.

**Conflicts of Interest:** The authors declare no conflict of interest.

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
