# Peer review of "Application of Macrophytes to the Assessment and Classification of Ecological Status above and below the Barrage with Hydroelectric Buildings"

_water, doi:10.3390/w11051028_

Round 1

Reviewer 1 Report

With ever mounting evidence of human impacts on the natural environment, globally, it is imperative to more fully document how consistently ( if not accurately) we are assessing the status of our natural environment across large geographic and national domains. This is particularly true for the hydrosphere and major aquatic natural systems near areas of human occupation.

The research reported here provides an assessment of the ecological status of three locations above, at, and below a barrage for a hydroelectric plant using the Polish assessment (MAMOR method) in greater detail, and three other EU methods from Ireland, Germany, and Bulgaria.

The research report is interesting and informative, in so far as the specific objectives that are designated.

However, I have some suggestions to be considered:

1)   Why did the authors select the particular methods used, beyond the MAMOR that would be particularly relevant to the nation where the study site was located? That is, what purpose or criteria were used in selecting the other three?

2)   As the authors point out based on overall assessment, the three methods give considerably consistent results. However, the RI-BG as the authors comment shows somewhat of a different profile. It would be helpful if the authors could briefly augment the Discussion by more fully addressing what aspect of the RI-BG might have contributed to these differences. And if appropriate, whether it is more likely to be biased (relative to the other three) for some ecosystems analyzed, compared to others, at least within the frame of the current selected sites near this hydroelectric barrage.

3)   While I agree with the authors’ assertion that the statistics chosen are applicable for this kind of research, it is fairly obvious that the data in Figure 5, for example, are so nearly identical,  and the number of groups analyzed (df = 3) so small, it is highly unlikely to realize a statistically significant result. The authors may want to at least acknowledge this. Also, declaring that the null hypothesis is not rejected, of course does not mean that the data are necessarily equivalent. The N and power of the test may not be sufficient. However, again, here the prima facie evidence is clear, that overall the results are highly similar. With df = 3 and a range in values of say 1 to 10, the scores for the four groups would have to be highly different to find a statistically significant difference.

4)   There are a few corrections in the text (that otherwise is nicely prepared and well organized): 

Line 105. Be more clear that this is still prior research. For example, “The prior published research that was carried out reveals both -----.” It is simply to be certain the reader understands that the research reported is not in some way partly summarized from the current research being reported here.

Line 510 The figure legend for Figure 5 needs to be expanded to explain the box graphs, what does the horizontal line represent and what does the diamond icon represent in each box of the diagrams ?

Line 167. “Taxon weight coefficient” not “Taxon weigh coefficient”.

Line 402. A note should be added in the title or at the bottom of the tables defining the column abbreviations:  P, P%, L, W, PLW, PW. I did not find these easily defined in the text.

Generally, throughout the text, the authors need to carefully check that all taxon binomial names of species are italicized, in some places they are not.

Be consistent to capitalize Table (not table) and Figure (not figure) in the text when referring to numbered figures and tables within the manuscript text.

Figure 5 is a helpful addition for the reader.

Author Response

Thank you very much for the substantive review of high scientific value. The remarks contained therein are valuable to us and we will use them for further scientific consideration. We would like to comment on the considerations contained in the review:

1) These methods were chosen in the article because they are considered as reference in the European Union countries in accordance with the provisions of the Water Framework Directive. They have been calibrated and harmonized in accordance with the methodology established on the basis of developed provisions on the EU scale (so-called intercalibration methods - this task is dealt with by Geographical International Groups = GIG, separated for individual regions - Central Baltic GIG deals with countries included in the intercalibration article; European Union has 37 reference methods for macrophytes, including 18 for rivers; EU methods database: http://www.wiser.eu/results/method-database/index.php?). Due to the fact that only groups of macrophyte assessment methods based on trophic indicators are used in the EU, it was decided to choose methods that differ in the methodology of conduct and emphasize other elements of the ecological status assessment. The Polish MMOR method draws attention to the degree of coverage with taxa, their index values and weighting factors, in the Irish MTR method – taxa and environmental tolerance of taxa, in the German TIM method – calculation of reactive phosphorus contained in plant tissues, while in Bulgarian RI-BG – an abundance of taxa and number of taxa in separate assessment groups. Despite the common denominator, i.e. the trophic evaluation, there are differences in the scientific approach within the methods, hence they could be compared with each other.

(Furse M. et al., The ecological status of European rivers: Evaluation and intercalibration of assessment methods, Hydrobiologia, 2006, 566: 3-29.

Schneider S., Macrophyte trophic indicator values from a European perspective. Limnologica, 2007, 37(4): 281-289)

2) Differences in the results obtained by the Bulgarian RI-BG method may differ due to the fact that the separated groups of methods are adapted to the river ecosystems present in the catchments of Bulgarian rivers (including the Danube river basin ecosystems). This is shown in Table 2, which describes the types of river groups in the method. The other methods are based on the types of abiotic rivers that are identical with each other (qualification as a sandy-loamy lowland river). It is true that the results obtained using the method used in Bulgaria may be burdened with a certain error resulting from greater bias for some of the analyzed ecosystems, as the Reviewer rightly pointed out.

(Gecheva G. & Yurukova L. D. Reference Aquatic Macrophyte Communities at Rivers in Southwestern Bulgaria. Comptes rendus de l'Académie bulgare des sciences: sciences mathématiques et naturelles, 2013, 66(66).

Pall K. et al. Bulgaria – report on fitting a classification method to the results of the completed intercalibration of the MedGIG (R-M1 and R-M2). BQE: Macrophytes. Ministry of Environment and Water, Bulgaria; Consortium DICON – UBA, 2016.)

3) Non-parametric significance tests (Wilcoxon rank sum test) were obtained using PROC NPAR1WAY. Assuming the test, there were no restrictions as to the size of the research groups. The dependent variables in the test were a water quality class obtained in three places using each method. This results in low df analysis, as the Reviewer rightly pointed out. Before the test, we analyzed other possibilities of comparing different methods. As it turned out, the test applied best reflected the nature of the sample. The test was presented on the graph presenting basic statistics. It seems to us that the test used was the best-fitted and possible statistical tool at this stage to compare the results of water quality assessment obtained by various methods.

 (SAS Institute Inc. SAS/STAT® 14.1 User’s Guide – Chapter 47: The NPAR1WAY Procedure. Cary, 2015, NC: SAS Institute Inc.)

4) Thank you for editorial comments. All the indicated errors have been corrected, which certainly increased the value of the article, which has become more readable, comprehensible and friendly in reception.

Reviewer 2 Report

The article provides a wealth of information on comparing both macrophyte methods and the impact of hydropower station on macrophyte communities. I definitely recommend it for publishing.
Based on this research, as a summary of his studies, the negative impact of hydropower station on macrophyte communities should be emphasized. eg. Based on our study marked the negative effects of hydroelectric buildings. All of the comparable macrophyte methodologies proved that the section above the reference location had become a poor water quality over the hydropower station, and the section below the hydropower station had not reached the reference state.
I would like to initiate the appearance of the mentioned above in the “Conclusions” chapter, which, in my opinion, would increase the value of the manuscript even more.

Author Response

Thank you very much for such a positive review. It is a great honor for us, and your suggestion has increased the substantive value of the manuscript even more. We are really grateful for such a positive response from you. Attached, we send a corrected version of the article containing the sentence suggested by you included in the conclusions.
